# The impact of organizational culture on professional fulfillment and burnout in an academic department of medicine

**Karen E. A. Burns**[1,2], **Reena Pattani**[1,2], **Edmund Lorens**[1], **Sharon E. Straus**[1,2], **Gillian A. Hawker**[1,3]*

**1** Department of Medicine, University of Toronto, Toronto, Ontario, Canada, **2** St. Michael's Hospital, Unity Health Toronto, Toronto, Ontario, Canada, **3** Women's College Research Institute, Women's College Hospital, Toronto, Ontario, Canada

* g.hawker@utoronto.ca

**Data Availability Statement:** Due to ethical restrictions by the University of Toronto, the data underlying this study is available upon request from the following contacts: Daniel Gyewu

## Abstract

Physician wellness is vital to career satisfaction, provision of high quality patient care, and the successful education of the next generation of physicians. Despite this, the number of physicians experience symptoms of burnout is rising. To assess the impact of organizational culture on physicians' professional fulfillment and burnout, we surveyed full-time Department of Medicine members at the University of Toronto. A cross-sectional survey assessed: physician factors (age, gender, minority status, disability, desire to reduce clinical workload); workplace culture (efforts to create a collegial environment, respectful/civil interactions, confidence to address unprofessionalism without reprisal, witnessed and/or personally experienced unprofessionalism); professional fulfillment and burnout using the Stanford Professional Fulfillment Index. We used multivariable linear regression to examine the relationship of measures of workplace culture on professional fulfillment and burnout (scores 0–10), controlling for physician factors. Of 419 respondents (52.0% response rate), we included 400 with complete professional fulfillment and burnout data in analyses (60% ≤ age 50, 45% female). Mean scores for professional fulfillment and burnout were 6.7±1.9 and 2.8±1.9, respectively. Controlling for physician factors, professional fulfillment was associated with satisfaction with efforts to create a collegial environment (adjusted beta 0.45, 95% CI 0.21 to 0.70) and agreement that colleagues were respectful/civil (adjusted beta 0.85, 95% CI 0.53 to 1.17). Lower professional fulfillment was associated with higher burnout scores. Controlling for professional fulfillment and physician factors, lower confidence in taking action to address unprofessionalism (adjusted beta -0.22, 95% CI -0.40 to -0.03) was associated with burnout. Organizational culture and physician factors had an impact on professional fulfillment and burnout. Professional fulfillment partially mediated the relationship between organizational culture and burnout. Strategies that promote inclusion, respect and civility, and safe ways to report workplace unprofessionalism are needed in academic medicine.

Research Ethics Manager, Health Sciences
Research Oversight & Compliance Office (ROCO)
(416) 978-3165 d.gyewu@utoronto.ca General
Inquiries: Human Research & Ethics Unit (HREU)
Research Oversight & Compliance Office (ROCO)
(416) 946-3273 ethics.review@utoronto.ca.

**Funding:** The authors received no specific funding
for this work.

**Competing interests:** Gillian Hawker receives
salary support as the Sir John and Lady Eaton
Professor and Chair of Medicine, Department of
Medicine, University of Toronto. Sharon Straus
holds a Tier 1 Canada Research Chair from the
Canadian Institutes of Health. Karen Burns holds a
Physician Services Incorporated Mid-Career
Clinical Research Award. No other competing
interests are declared. We confirm that salary
support and personnel awards did not impact our
ability to adhere to PLOS ONE policies on data
sharing and materials.

# Introduction

Physician wellness is vital to career satisfaction, provision of high quality patient care, and the successful education of the next generation of physicians [1–11]. Despite this, research supports that almost half of all physicians experience burnout symptoms [12–14], characterized by work-related emotional exhaustion, depersonalization, and a sense of reduced personal accomplishment [15]. Furthermore, the rates of burnout among physicians are rising [15].

Healthcare organizational factors, including excessive workloads, workflow inefficiencies, increased time spent in documentation, lack of input or control, and loss of meaning at work, contribute to physician burnout [16]. Among medical residents, unprofessional workplace behavior, and specifically intimidation, harassment and discrimination, have also been linked to lower work satisfaction and higher burnout [17]. A cross-sectional survey of 300 non-medical university faculty members found that both direct and indirect forms of harassment (gender, sexual, and academic) were higher in female than male faculty and associated with higher burnout, as measured by the Maslach Burnout Inventory (MBI) [18, 19]. Whether workplace unprofessionalism independently contributes to burnout among academic physicians is unknown.

We examined the relationship of organizational culture to professional fulfillment and burnout among full-time clinical faculty members in a large academic Department of Medicine (DOM). We hypothesized that, after controlling for factors linked to higher rates of physician burnout, organization culture would influence professional fulfillment and burnout.

# Materials and methods

## Study design and participants

We conducted a cross-sectional survey of full-time clinical faculty members in the DOM at the University of Toronto in 2019. Full-time clinical faculty members hold a primary clinical appointment at one of six university-affiliated hospitals; each hospital has a department of medicine and physician-in-chief. The study was approved by the Research Ethics Board of the University of Toronto. Participants provided informed consent.

## Questionnaire design, testing and administration

Members of the DOM Mentorship, Equity and Diversity Committee designed, pilot-tested, and administered the questionnaire (S1 File) anonymously using SurveyMonkey (SVMK, San Mateo, USA) [20, 21]. A pre-notification e-mail was sent by the University DOM Chair prior to a second e-mail, bearing a unique survey link to each potential respondent. Reminder emails were sent at two-week intervals to maximize response rate. An honest broker (EL) collated responses to preserve respondent anonymity.

## Assessments

We collected data to characterize respondents' socio-demographic and professional characteristics. Socio-demographic variables included age (≤50, 51–60, 61–70, >70 years); gender (female/male); self-reported disability status (yes/no); under-represented minority (URM) membership (yes/no); socioeconomic status (SES) as a child/adolescent (lower/lower-middle/middle/upper-middle/upper); academic rank (lecturer/assistant professor/associate professor/professor), clinical specialty and academic position to determine protected time for scholarly work. Participants were asked whether they were interested in reducing their clinical workload to aid work-life integration (yes/no/not sure). To assess organizational culture, we asked participants to report their level of satisfaction with efforts made to create a collegial/supportive/inclusive environment

(5-point Likert scale from 'strongly satisfied' to 'strongly dissatisfied'); level of agreement that colleagues interact with them in a respectful/civil manner (5-point Likert from 'strongly disagree' to 'strongly agree'); and level of confidence that they could take action to address unprofessionalism without reprisal (5-point Likert from 'very doubtful' to 'strongly confident') in each of the University DOM, University Division, Primary Hospital DOM, and Primary Hospital Division. Respondents were asked if they had witnessed unprofessionalism (e.g., disrespect, abuse, bullying, micro-aggression, or discrimination) by faculty members towards others in the last two years (yes/no) and whether they had personally experienced these behaviours (yes/no). We used the Stanford Professional Fulfillment Index (PFI) to assess professional fulfillment and burnout [22]. The Stanford PFI is composed of three subscales: 6-item professional fulfillment subscale; 4-item work exhaustion subscale; and 6-item interpersonal disengagement subscale. Subscales are scored from 0–10 with higher professional fulfillment and lower work exhaustion and interpersonal disengagement scores representing more favorable responses. Scores for work exhaustion and interpersonal disengagement were combined to assess burnout (score 0–10) with higher scores indicate more burnout symptoms and scores $\geq$ 3.3 indicating burnout [22].

Our **outcomes of interest** were Stanford PFI scores for professional fulfillment and burnout. **Exposures of interest** were the 4 measures of organizational culture (collegial environment; respectful/civil interactions with colleagues; confidence in addressing unprofessionalism; and witnessed and experienced unprofessionalism). **Covariates of interest** were physician factors (age group, gender, URM status, disability, SES background, academic rank, specialty, and clinical workload).

## Statistical analysis

We summarized binary and continuous data using proportions and means and medians, respectively. We assessed sample representativeness by comparing respondent characteristics to those of the sampling frame. We combined responses regarding organizational culture across the University and hospital DOMs and University and hospital Hospital Divisions into a single 10-point scale. We categorized protected time for scholarly work as 'low/moderate/high' for academic position descriptions with ≤30%, 31–69%, and 70%+ protected time, respectively. We report data elements with 6 or more responses.

We conducted multivariable linear regression analyses to examine associations between exposures of interest and each of professional fulfillment and burnout, before and after controlling for covariates [23]. To test the hypothesis that the relationship between organizational culture and burnout was mediated, part, by respondents' professional fulfillment, we examined the effect on the relationship between measures of organizational culture and burnout after controlling for professional fulfillment. We assessed collinearity of independent variables using a Variance Inflation Factor of >4 [24]. We performed statistical analyses using SPSS 26.0 (IBM SPSS Statistics for Windows, Armonk, NY: IBM Corp) and SAS Version 9.4 (SAS Institute Inc., Cary, NC). P-values <0.05 were considered significant.

## Results

### Respondent characteristics

Of 805 eligible faculty members, 419 (52.1%) completed the questionnaire. Of these, we included 400 questionnaires with complete Stanford PFI data. Respondents were similar to non-respondents with respect to age group and hospital affiliation but were significantly less likely to be a man (60.6% vs. 51.0%), assistant professor (43.6% vs. 38.2%) or cardiologist (14.4% vs. 9.2%), respectively. Of respondents, 192/320 (60.0%) were 50 years of age or

younger, 153/340 (45.0%), female, 109/366 (29.8%), URM, 78/366 (21.3%), or from a lower/lower-middle SES background. Twelve respondents [12/352 (3.4%)] reported a disability (Table 1). Based on academic position descriptions, 96/337 (28.5%) had high protected time for scholarly work, while 117 (34.7%) and 124 (36.8%) had moderate or little protected time, respectively. Nearly half of respondents [174/380 (45.8%)] wanted to reduce their clinical workload.

## Perceptions of organizational culture

On average, mean agreement was high that colleagues treated them respectfully/civilly (mean score 8.4±1.9) (Table 2). Mean scores for level of satisfaction with efforts made by the institution to create a collegial environment and level of confidence in addressing unprofessional behavior by colleagues without fear of retaliation were lower (6.5±2.4 and 6.1±2.8, respectively) Several respondents indicated that they personally experienced unprofessionalism (41.0%) or witnessed unprofessional behaviours by faculty members towards others (without personally experiencing them) (18.8%) within the last 2 years. Unprofessional behaviours most frequently included disrespect (35.8%), micro-aggressions (20.5%), and bullying (17.3%).

## Self-reported professional fulfillment and burnout

Most respondents (85.4%) found their work meaningful. Approximately two-thirds felt worthwhile at work (64.3%), found their work satisfying (71.2%), and felt they were contributing professionally in ways they valued (70.8%). Although half (52.8%) affirmed that they felt happy at work, only 37.1% felt in control when dealing with difficult problems at work. About half of respondents reported feeling physically exhausted at work (56.3%), a sense of dread when they thought about the work they had to do (49.5%) and emotionally exhausted at work (46.0%). Some (30.5%) respondents reported lacking enthusiasm at work, while 18.1% reported feeling at least moderately less interested in talking to patients and 20.4% reported being at least moderately less sensitive to the feelings of others. Over a quarter of physicians affirmed feeling less empathetic and connected with colleagues (28.5% and 33.1%, respectively) and 16.3% and 18.3% of physicians affirmed feeling less empathetic and connected with patients, respectively. Mean scores for professional fulfillment, emotional exhaustion and interpersonal disengagement were 6.7±1.9, 3.6±2.2, and 2.2±1.9, respectively. The mean score for burnout was 2.8±1.9 with 31.8% of respondents meeting burnout criteria.

## The relationship of organizational culture to professional fulfillment

In univariate analyses, positive perceptions of the organizational culture and higher academic rank were associated with greater professional fulfillment. Conversely, having personally experienced unprofessionalism at work, female gender, self-reported disability, low or moderate protected time for scholarly work, and a desire to reduce clinical workload were associated with lower scores (Table 3). In multivariable modeling, professional fulfillment was significantly higher among those with greater satisfaction with efforts to encourage a supportive/collegial workplace (adjusted beta per level of satisfaction 0.45, 95% CI 0.21 to 0.70) and greater agreement that colleagues were respectful/civil (adjusted beta per level of agreement 0.85, 95% CI 0.53 to 1.17) and significantly lower among respondents with self-reported disability (adjusted beta -1.33, 95% CI -2.22 to -0.44), but unrelated to confidence in taking action regarding unprofessional behavior without fear of retaliation or unprofessional behaviors.

**Table 1. Characteristics of physician respondents (n = 400).**

| Respondent Characteristic | N (%) |
|---|---|
| **Age group** | |
| ≤50 | 192/320 (60.0) |
| 51–60 | 73 (22.8) |
| 61–70 | 45 (14.1) |
| >70 years | 10 (3.1) |
| **Gender** | |
| Woman | 153 (45.0) |
| Man | 187 (55.0) |
| **Under-Represented Minority** | 109/366 (29.8%) |
| **Socioeconomic Background** | |
| Low/Low-Middle | 78/366 (22.5) |
| Middle | 135 (39.0) |
| Upper middle/Upper | 133 (38.4) |
| **Self-reported disability** | 12/352 (3.4) |
| **Medical Specialty** | |
| Cardiology | 33/321 (10.3) |
| Critical Care | 15 (4.7) |
| Emergency Medicine | 26 (8.1) |
| Endocrinology | 25 (7.8) |
| Gastroenterology | 16 (5.0) |
| General Internal Medicine | 41 (12.8) |
| Geriatrics | 11 (3.4) |
| Hematology | 26 (8.1) |
| Infectious Diseases | 14 (4.4) |
| Medical Oncology | 19 (5.9) |
| Nephrology | 11 (3.4) |
| Neurology | 22 (6.9) |
| Physiatry | 11 (3.4) |
| Respirology | 21 (6.5) |
| Rheumatology | 20 (6.2) |
| Other[a] | 10 (3.1) |
| **Protected time for scholarly work**[b] | |
| 70%+ | 96/337 (28.5) |
| 40–50% | 117 (34.7) |
| 20–30% | 124 (36.8) |
| **Academic rank** | |
| Lecturer | 12/335 (3.6) |
| Assistant Professor | 139 (41.5) |
| Associate Professor | 79 (23.6) |
| Full Professor | 105 (31.3) |

[a]Includes Dermatology, Clinical Allergy & Immunology, Occupational Medicine, Palliative Medicine & Clinical Pharmacology & Toxicology.

[b]Clinician scientists and administrators have ≥ 70% time protected for non-clinical work, Clinician investigators and Clinician educators have ~ 50% of their time protected for non-clinical work, and Clinician teachers and Clinicians in quality and innovation have 20–30% of their time protected for non-clinical work.

**Table 2. Physicians' perceptions of the organizational culture.**

| Measure | |
|---|---|
| Level of satisfaction with efforts to create a collegial environment–mean (SD) | 6.5 (2.4) |
| Level of agreement that colleagues are respectful & civil–mean (SD) | 8.4 (1.9) |
| Level of confidence addressing unprofessionalism without retaliation–mean (SD) | 6.1 (2.8) |
| Unprofessionalism–n (%) | |
| Witnessed but not experienced | 73 (18.8) |
| Personally experienced | 159 (41.0) |
| Stanford Index–mean (SD) | |
| Professional Fulfillment | 6.7 (1.9) |
| Work Exhaustion | 3.6 (2.2) |
| Interpersonal Disengagement | 2.2 (1.9) |
| Burnout | 2.8 (1.9) |
| Interest in reducing clinical workload to improve work-life balance–n (%) | |
| Yes | 174/380 (45.8) |
| No | 133 (35.0) |
| Uncertain | 73 (19.2) |

## The relationship of organizational culture to correlates of burnout

In univariate analyses, greater burnout symptoms were associated with less positive perceptions of organizational culture and personal experiences with unprofessionalism at work in addition to younger age, female gender, URM status, self-reported disability, lower rank, less protected time for scholarly work, and a desire to reduce clinical workload (Table 4). In multivariable modeling, greater burnout symptoms were associated with younger age, self-reported disability (parameter estimate 1.55, 95% CI 0.49 to 2.61), interest in reducing clinical workload (parameter estimate 1.20, 95% CI 0.77 to 1.63), lower satisfaction with efforts to encourage a supportive/collegial workplace (adjusted beta per unit increase in satisfaction -0.32, 95% CI -0.57 to -0.07), lower agreement that colleagues were respectful/ civil (adjusted beta per unit increase in agreement -0.49, 95% CI -0.81 to -0.16), and lower confidence in taking action regarding unprofessional behavior without fear of retaliation (adjusted beta per unit increase in confidence -0.31, 95% CI -0.51 to -0.10). When professional fulfillment was added to the multivariable model, fulfillment was inversely associated with burnout (adjusted beta per point increase in Stanford PFI score -0.45, 95% CI -0.56 to -0.33). Although, the effects of satisfaction with efforts to encourage a supportive/collegial workplace and agreement that colleagues were respectful/civil on burnout were attenuated and became non-significant, declining confidence in taking action regarding unprofessional behavior remained significantly associated with greater burnout symptoms (adjusted beta per unit increase in confidence -0.22, 95% CI -0.40 to -0.03) (Table 5).

## Discussion

We examined the relationship between measures of the organizational culture and professional fulfillment and burnout. Controlling for identified risk factors for burnout, including gender, career stage, and clinical workload, we found that physicians' satisfaction with efforts made to encourage a supportive/collegial workplace and level of agreement that colleagues were respectful/civil were significant contributors to professional fulfillment. Greater professional fulfillment and confidence in taking action to address unprofessional behavior without fear of retaliation were associated with fewer burnout symptoms. Controlling for these factors,

**Table 3. Relationship of physician and organization factors to professional fulfillment–results of linear regression modeling.**

| Independent Variable | Dependent Variable: Stanford Professional Fulfillment Score | | | | |
|---|---|---|---|---|---|
| | Unadjusted Parameter Estimate | 95% Confidence Limits | | Adjusted Parameter Estimate | 95% Confidence Limits | |
| **Physician Factors** | | | | | | |
| Age (ref ≤ 50 years) | | | | | | |
| 51–60 | 0.49 | -0.02 | 1.01 | 0.19 | -0.30 | 0.69 |
| 61–70 | 0.57 | -0.05 | 1.19 | 0.68 | -0.02 | 1.37 |
| 71 + | 1.13 | -0.04 | 2.30 | 0.83 | -0.25 | 1.90 |
| Female gender (ref = male) | -0.51 | -0.92 | -0.10 | -0.13 | -0.51 | 0.24 |
| URM[a] (ref = no) | -0.38 | -0.81 | 0.06 | 0.10 | -0.30 | 0.49 |
| SES background (ref = lower/middle) | | | | | | |
| Middle/Upper-middle | 0.36 | -0.14 | 0.85 | 0.07 | -0.36 | 0.50 |
| Upper | 0.23 | -0.88 | 1.35 | -0.24 | -1.16 | 0.68 |
| Self-reported disability (ref = no) | -1.65 | -2.72 | -0.57 | -1.33 | -2.22 | -0.44 |
| Current academic rank (ref = Lecturer/Assistant Professor) | | | | | | |
| Associate Professor | 0.70 | 0.19 | 1.20 | 0.32 | -0.17 | 0.81 |
| Professor | 1.14 | 0.68 | 1.60 | 0.27 | -0.34 | 0.87 |
| Protected time for scholarly work (ref = a lot) | | | | | | |
| Little (low) | -0.86 | -1.36 | -0.35 | -0.04 | -0.53 | 0.45 |
| Some (moderate) | -0.55 | -1.07 | -0.04 | 0.08 | -0.38 | 0.53 |
| **Workplace Culture Factors** | | | | | | |
| Collegial & Supportive–per unit increase | 0.45 | 0.20 | 0.70 | 0.45 | 0.21 | 0.70 |
| Respectful & Civil–per unit increase | 0.69 | 0.39 | 0.99 | 0.85 | 0.53 | 1.17 |
| Confidence take action–per unit increase | 0.41 | 0.23 | 0.59 | 0.20 | 0.00 | 0.41 |
| Unprofessional behavior in workplace (ref = none) | | | | | | |
| Witnessed but not personally experienced | -0.36 | -0.92 | 0.19 | -0.20 | -0.69 | 0.29 |
| Personally experienced | -1.06 | -1.50 | -0.63 | -0.05 | -0.49 | 0.38 |
| Desires reduction in clinical workload (ref = no) | -1.01 | -1.45 | -0.57 | -0.42 | -0.85 | 0.00 |

[a]URM = under-represented minority.

we found no relationship between having witnessed or personally experienced unprofessional behavior with either professional fulfillment or burnout. These findings suggest a need to develop and implement strategies that promote inclusion, respect and civility, and safe ways to identify and act upon workplace unprofessionalism in academic medicine.

This is the first study to examine the relationship between organizational culture and both professional fulfillment and burnout. Both fulfillment and burnout were associated with physicians' perceptions regarding efforts made by our DOM to promote a diverse and inclusive workplace and the level of workplace respect/civility, while confidence in addressing unprofessional behavior without fear of retaliation was correlated with a lower risk of burnout. Controlling for these factors, physicians' reported experiences of workplace unprofessionalism by colleagues were unrelated to both professional fulfillment and burnout. This suggests that an organization's expectations for professional behavior and response to unprofessionalism, is an important buffer against the negative effects of microaggressions and unprofessional behaviours/discourse. These findings suggest a need for interventions to enhance workplace professionalism and enable safe reporting of incivility [25].

Physician burnout has been linked to female gender, URM status, earlier career stage (age/rank), excessive clerical and regulatory workload, and lack of control at work [5, 16, 26–30]. In

**Table 4. Relationship of physician and organization factors to symptoms of burnout–results of linear regression modeling.**

| Independent Variable | Dependent Variable: Stanford Burnout Score | | | | |
|---|---|---|---|---|---|
| | Unadjusted Parameter Estimate | 95% Confidence Limits | | Adjusted Parameter Estimate | 95% Confidence Limits | |
| **Physician Factors** | | | | | | |
| Age (ref = <50 years) | | | | | | |
| 51–60 | -0.57 | -1.07 | -0.06 | -0.14 | -0.65 | 0.36 |
| 61–70 | -1.2 | -1.83 | -0.63 | -0.97 | -1.68 | -0.27 |
| 71+ | -1.4 | -2.60 | -0.32 | -0.91 | -2.00 | 0.17 |
| Female gender (ref = male) | 0.85 | 0.45 | 1.24 | 0.24 | -0.14 | 0.61 |
| URM[a] (ref = no) | 0.60 | 0.17 | 1.03 | -0.105 | -0.55 | 0.26 |
| SES background (ref = lower/middle) | | | | | | |
| Middle/Upper-middle | -0.32 | -0.81 | 0.17 | -0.07 | -0.50 | 0.37 |
| Upper | 0.08 | -1.02 | 1.17 | 0.40 | -0.53 | 1.34 |
| Self-reported disability (ref = no) | 1.55 | 0.49 | 2.61 | 1.29 | 0.39 | 2.19 |
| Current academic rank (ref = Lecturer or Assistant Professor) | | | | | | |
| Associate Professor | -0.52 | -1.01 | -0.02 | -0.29 | -0.79 | 0.20 |
| Professor | -1.35 | -1.80 | -0.90 | -0.21 | -0.83 | 0.40 |
| Protected time for scholarly work (ref = a lot) | | | | | | |
| Little (low) | 0.91 | 0.41 | 1.41 | 0.12 | -0.38 | 0.62 |
| Some (moderate) | 0.42 | -0.08 | 0.93 | -0.10 | -0.56 | 0.36 |
| **Workplace Culture Factors** | | | | | | |
| Collegial & Supportive–per unit increase | -0.31 | -0.57 | -0.06 | -0.32 | -0.57 | -0.07 |
| Respectful & Civil–per unit increase | -0.34 | -0.66 | -0.03 | -0.49 | -0.81 | -0.16 |
| Confidence take action–per unit increase | -0.60 | -0.79 | -0.42 | -0.31 | -0.51 | -0.10 |
| Unprofessional behavior in workplace (ref = none) | | | | | | |
| Witnessed but not personally experienced | 0.51 | -0.02 | 1.04 | 0.30 | -0.20 | 0.79 |
| Personally experienced | 1.41 | 0.99 | 1.82 | 0.36 | -0.08 | 0.80 |
| Desires reduction in clinical workload (ref = no) | 1.20 | 0.77 | 1.63 | 0.62 | 0.19 | 1.05 |

[a]URM = under-represented minority.

a study comparing the predilection for burnout in physicians vs. individuals with a graduate or professional degree, burnout symptoms were significantly more common in physicians and persisted despite adjustment for age, sex, relationship status, and hours worked/week in multi-variable analysis [31]. Although each of these factors was associated with greater burnout symptoms in our univariate analyses, these relationships were attenuated after controlling for measures of organizational culture [32–34]. One interpretation of these findings is that younger, female, and URM status faculty may be more likely to experience burnout due to differences in their perceptions or experiences of workplace culture.

Although limited to 12 respondents, we found a relationship between having a disability and both lower professional fulfillment and higher burnout scores. To our knowledge no prior study has reported these relationships. Unlike gender and URM status, the relationships between disability and both professional fulfillment and burnout were not attenuated by factors related to organizational culture suggesting that the effect of disability on both metrics may be through different mechanisms related to environmental barriers [35] or the effort required by disabled physicians to complete work-related tasks. Additional research is needed to confirm these findings in disabled physicians and if true to ascertain how more inclusive work environments can be created for disabled physicians.

Table 5. Correlates of burnout–professional fulfillment added to the multivariable model[a].

| Independent Variable | Dependent Variable: Stanford Burnout Score | | | | |
|---|---|---|---|---|---|
| | Adjusted Parameter Estimate | 95% Confidence Limits | | Adjusted Parameter Estimate | 95% Confidence Limits | |
| **Physician Factors** | | | | | | |
| Age (ref = <50 years) | | | | | | |
| 51–60 | -0.14 | -0.65 | 0.36 | -0.055 | -0.51 | 0.40 |
| 61–70 | -0.97 | -1.68 | -0.27 | -0.67 | -1.31 | -0.03 |
| 71+ | -0.91 | -2.00 | 0.17 | -0.545 | -1.53 | 0.44 |
| Self-reported disability (ref = no) | 1.29 | 0.39 | 2.19 | 0.69 | -0.13 | 1.52 |
| **Workplace Culture Factors** | | | | | | |
| Collegial & Supportive–per unit increase | -0.32 | -0.57 | -0.07 | -0.12 | -0.35 | 0.115 |
| Respectful & Civil–per unit increase | -0.49 | -0.81 | -0.16 | -0.11 | -0.41 | 0.20 |
| Confidence take action–per unit increase | -0.31 | -0.51 | -0.10 | -0.22 | -0.40 | -0.03 |
| Desires reduction in clinical workload (ref = no) | 0.62 | 0.19 | 1.05 | 0.43 | 0.04 | 0.82 |
| **Professional Fulfillment–per unit increase in score** | | | | -0.45 | -0.56 | -0.33 |
| Adjusted $R^2$ | 0.378 | | | 0.497 | | |

[a]Controlling additionally for gender, under-represented minority (URM) status, socioeconomic status (SES) background, rank, protected time, witnessed and personally experienced unprofessional behavior.

Large-scale surveys in the United States have reported physician burnout rates ranging from 43.9%-54.4% [15, 36, 37]. Conversely, only 31.8% of our physicians met criteria for burnout. This may reflect respondent bias as a lower response rate among assistant professors would bias towards a lower burnout rate. Conversely, a lower response rate among male faculty would bias towards a higher burnout rate. Lower burnout rates may also reflect different practice contexts (Canadian Universities and hospitals, socialized medicine, favorable medicolegal climate) or the instruments used to assess burnout. A study that compared the performance characteristics of the Stanford PFI to the MBI, found a moderate correlation between the Stanford PFI burnout measures with their closest related MBI equivalents (r≥0.50) [22].

Our study has several strengths. First, it is the largest survey conducted to date using the Stanford PFI, which was specifically designed to assess work-related well-being among physicians. The Stanford PFI [22] has three key advantages compared to the MBI [19]. It concurrently assesses professional fulfillment and burnout, assesses interpersonal disengagement and burnout in interactions with patients and colleagues, and evaluates metrics over a two-week time horizon—reducing the potential for recall bias. Second, we sampled academic physicians across multiple hospitals, specialties, academic foci, and stages of academic career development. Third, our study is novel in assessing the impact of features of organizational culture, as opposed to organizational factors (workflow, workload, time spent in documentation, lack of control, and loss of meaning), on professional fulfillment and burnout. Finally, our response rate aligns with those of prior multi-disciplinary, cross-sectional surveys of physicians [38, 39]. Our study also has limitations. Gender was evaluated as binary. Survey respondents differed marginally from the DOM membership with respect to gender and specialty [40]. Notwithstanding, the relationships that we identified are valid within our sample but require confirmation in other cohorts and contexts. To this end, our findings may not be generalizable to non-respondents, other universities/departments, and settings.

Given the implications of burnout for physicians, patients, and healthcare systems [41–44], our findings indicate that there is a need for greater emphasis on culture in academic medicine, specifically with respect to promoting inclusion, respect and civility, and safe ways to

report workplace unprofessionalism. They also highlight the importance of an institutional commitment to creating a positive work environment and the need for a fair and transparent reporting process to address workplace unprofessionalism. To this end, our DOM developed and implemented strategies to address organizational culture at the University of Toronto including education (cultural sensitivity, allyship, and implicit bias); changes in organizational structure, policies, and processes (fairness and transparency of appointments, addressing workplace incivility) and mentorship [45]. The extent to which physician wellbeing may be improved by efforts to address equity, diversity, inclusion, and organizational culture (professionalism, and mechanisms to report and address unprofessional behaviours) remains to be fully elucidated.

Physicians' perceptions of the organizational culture at work were strongly related to their self-reported professional fulfillment and burnout. Strategies that promote inclusion, respect and civility, and safe ways to report workplace unprofessionalism are needed in academic medicine.

## Supporting information

**S1 File. Department of medicine faculty survey 2019.**
(PDF)

## Acknowledgments

The authors would like to thank the following individuals for their contributions to this work: **Members of the inaugural Department of Medicine Mentorship, Equity and Diversity Committee**: Chair—Sharon Straus; Interim Co-Chairs—Andrea Page and Caroline Chessex; Members At Large—Anita Balakrishna, Glen Bandiera, Lilian Belknap, Mary Bell, Karen Burns, Nora Cullen, Loretta Daniel, Shiphra Ginsburg, Michael Gordon, Ayelet Kuper, Christie Lee, Liesly Lee, Heather McDonald-Blumer, Sangeeta Mehta, Sarah Meilach, Clare Mitchell, Danny Panisko, Reena Pattani, Shail Rawal, Lisa Richardson, Larry Robinson, Paula Rochon, Jarred Rosenberg, Sam Sabbah, Nazia Selzner, Malika Sharma, Michelle Silver, Kathryn Tinckam, Maureen Trudeau, Beryl Tsang, Katina Tzanetos, and Robert Wu; **Executives of the Department of Medicine**, and in particular the **Departmental Vice Chairs** (Drs. Michael Farkouh, Arno Kumagai, Philip Marsden and Kaveh Shojania), **Physicians-in-Chief of Medicine** (Drs. Chaim Bell, Edward Cole, Paula Harvey, Michelle Hladunewich, Kevin Imrie, Gary Naglie, Gary Newton, Tom Parker), and **Departmental Division Directors** (Drs. Anil Adiseh, Johane Allard, Claire Bombardier, Douglas Bradley, Laurent Brochard, Anil Chopra, Paul Dorian, Linn Holness, Stephen Hwang, Jacqueline James, David Juurlink, Moira Kapral, Rupert Kaul, Monika Kryzanowska, Anthony Lang, Gary Lewis, Barbara Liu, Susanna Mak, Heather McDonald-Blumer, Xavier Montalban, Isaac Odame, Rulan Parekh, Vincent Piquet, Kathy Pritchard, Larry Robinson, Neil Shear, Gordon Sussman, Laura Targownik, Peter Vadas, Camilla Zimmerman) who served in leadership roles over the study time period for their support and advocacy of gender equity in the department. This work would not have been accomplished without their leadership. **Ms. Jean Robertson**, Director of Human Resources, Faculty of Medicine, University of Toronto, who provided the non-DOM data to the authors for this study.

## Author Contributions

**Conceptualization:** Karen E. A. Burns, Reena Pattani, Sharon E. Straus, Gillian A. Hawker.

**Data curation:** Karen E. A. Burns, Reena Pattani, Edmund Lorens, Sharon E. Straus, Gillian A. Hawker.

**Formal analysis:** Edmund Lorens.

**Methodology:** Karen E. A. Burns, Reena Pattani, Sharon E. Straus, Gillian A. Hawker.

**Project administration:** Karen E. A. Burns, Reena Pattani.

**Resources:** Gillian A. Hawker.

**Supervision:** Karen E. A. Burns, Gillian A. Hawker.

**Writing – original draft:** Karen E. A. Burns, Gillian A. Hawker.

**Writing – review & editing:** Karen E. A. Burns, Reena Pattani, Edmund Lorens, Sharon E. Straus, Gillian A. Hawker.

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
