## [Decision Letter · Decision Letter 0]

15 Feb 2021

PONE-D-20-32481

The impact of organizational culture on professional fulfillment and burnout in an academic Department of Medicine

PLOS ONE

Dear Dr. Burns,

Thank you for submitting your manuscript to PLOS ONE. After careful consideration, we feel that it has merit but does not fully meet PLOS ONE’s publication criteria as it currently stands. Therefore, we invite you to submit a revised version of the manuscript that addresses the points raised during the review process.

We look forward to receiving your revised manuscript.

Kind regards,

Gerard Hutchinson, MD

Academic Editor

PLOS ONE

Journal Requirements:

2. Please include additional information regarding the survey or questionnaire used in the study and ensure that you have provided sufficient details that others could replicate the analyses.

For instance, if you developed a questionnaire as part of this study and it is not under a copyright more restrictive than CC-BY, please include a copy, in both the original language and English, as Supporting Information.

'The funders had no role in study design, data collection and analysis, decision to publish, or preparation of the manuscript.'

'Gillian Hawker receives salary support as the Sir John and Lady Eaton Professor and Chair of Medicine, Department of Medicine, University of Toronto. Sharon Straus holds a Tier 1 Canada Research Chair from the Canadian Institutes of Health. Karen Burns holds a Physician Services Incorporated Mid-Career Clinical Research Award. No other competing interests are declared.'

Reviewers' comments:

Reviewer's Responses to Questions

**Comments to the Author**

1. Is the manuscript technically sound, and do the data support the conclusions?

Reviewer #1: Yes

Reviewer #2: Yes

2. Has the statistical analysis been performed appropriately and rigorously? 

Reviewer #1: Yes

Reviewer #2: Yes

3. Have the authors made all data underlying the findings in their manuscript fully available?

Reviewer #1: Yes

Reviewer #2: No

4. Is the manuscript presented in an intelligible fashion and written in standard English?

Reviewer #1: Yes

Reviewer #2: Yes

5. Review Comments to the Author

Reviewer #1: It was a pleasure reviewing the manuscript “The impact of organizational culture on professional fulfilment and burnout in an academic Department of Medicine”

The topic is relevant and the idea innovative. Burnout is frequent in the ICU setting and it is largely undiagnosed. The mortality rate of adult ICU patients depends on the severity of illness and patient population; it ranges between 6.4% and 80% during the COVID-19 Pandemic. Interdisciplinary work between many medical professions is essential in an ICU to deliver high quality patient care. Members of the ICU team include physicians, nurses, technicians, therapists, nutritionists, pharmacists, and other support staff making effective management an important element of a highly functioning ICU. Currently, health care delivery has far lower levels of reliability than that achieved in other industries (e.g., aviation) and staff are not supported efficiently. Successful improvement of Critical Care requires a perspective that treats the ICU as a complex, socio-technical system and prevents staff to suffer burnout. ICU healthcare workers who provide aggressive care to critical patients have moral distress and are at risk for burnout, which in turn can lead to poor quality patient care and higher job turnover rates (Meltzer et al. Am J Crit Care 2004, Corley et al. J Adv Nurs 2000, Appropricus ESICM Abstract).

In another survey 70% of ICU workers perceived conflicts, (Azoulay E, Timsit JF, Sprung CL, et al. Prevalence and Factors of Intensive Care Unit Conflicts: The Conflicus Study. Am J Respir Crit Care Med. 2009 Jul 30m). These were usually considered harmful and were significantly associated with job strain. Workload, communication, and end-of-life care emerged as potential targets for improvement. Hamric and colleagues evaluated moral distress in both nurses and doctors. They found that ICU nurses experienced moral distress more often than physicians; however, similar situations provoked moral distress in both groups (Hamric et al. Crit Care Med 2007). Studies related to moral distress and futile or inappropriate care in the ICU do not provide patient-linked data. Consequently, the real extent of the problem of perceived inappropriate care at the end-of-life in the ICU is unknown and the magnitude of situations causing feelings of moral distress may be underestimated. Feelings like frustration, stress, guiltiness, lack of motivation, lack of communication, isolation and finally burn out are common. There is evidence in the medical and non-medical literature suggesting that the burn out leads to low performance and concentration. The performance of all Critical Care workers is prone to error and is often associated with monitoring and daily care tasks like ordering of medication or execution of patient treatment. ERRORS cause distress and can lead to more burnout. A blaming culture can lead to more mistakes and so on. The organisational culture is as complex to measure as is the incidence or severity of burnout. There are few validated tools and these are limited. This study is looking at a relevant element and it is using one those few tools that can provide reliable information. The Stanford Professional Fulfilment Index (PFI) reports on professional fulfilment, work exhaustion, interpersonal disengagement.

Major concerns:

The limit of most surveys is the response rate which in this occasion is just above 50%.

The target people for this survey are the faculty members, in reality the entire staff could be included. The ICU is a complex system and isolating one part of the staff can have benefits and disadvantages such as reporting only on one component or one side of the problem.

The percentage of people happy with their condition was high. Results presented in this manuscript would be more valuable if a benchmark was available.

This model could be used in several institutions and compared in future.

Minor concerns:

-small error in the text and the title page : “The impact of organizational culture on professional fulfillment and burnout in an academic…”

Please change fulfilment

Reviewer #2: Thank you for the opportunity to review this paper by Dr. Burns and colleagues on the impact of organizational culture on professional fulfillment and burnout in an academic department of medicine. This well-written paper reports the results of a cross-sectional survey of 400 full-time physicians. I have a few comments/queries for the authors to consider:

Abstract:

1) Please provide the score range for the SPFI so the mean scores presented in the results are interpretable

2) Are the "physician factors" referred to on line 40 the same as the "physician characteristic" defined on line 34? if so perhaps change one of the terms for consistency.

Introduction

4) The introduction is concise and lays out both the problem and gap well.

Methods

5) For the assessments, more detail on some of the definitions would be helpful. For example, for self-reported disability status, how was this question asked? Does it include physical, mental, or cognitive disabilities? For participant age, why were those grouping selected? 60% of respondents are in the <50 are group.

Results

6) The sentence starting on line 138 is confusing and it is difficult to tell which numbers apply to which groups given the placement of comas.

Discussion

7) The paragraph (starting line 251) discussing the novel findings related to disability should be tempered slightly. With only 12 people in this group the results should be interpreted with caution.

6. PLOS authors have the option to publish the peer review history of their article (what does this mean?). If published, this will include your full peer review and any attached files.

Reviewer #1: **Yes: **Francesca Rubulotta

Reviewer #2: No

---

## [Author Response · Author response to Decision Letter 0]

27 Apr 2021

ONE-D-20-32481

The impact of organizational culture on professional fulfillment and burnout in an academic Department of Medicine

Submitted by Dr. Karen E. A. Burns

Date: Feb 23, 2021

 Done. Thank you.

2. Please include additional information regarding the survey or questionnaire used in the study and ensure that you have provided sufficient details that others could replicate the analyses.For instance, if you developed a questionnaire as part of this study and it is not under a copyright more restrictive than CC-BY, please include a copy, in both the original language and English, as Supporting Information.

 We have appended the questionnaire as Appendix 1. Thank you for this suggestion.

Although, we have no difficulty sharing our database, we (Department of Medicine, under the leadership (Chair) of Dr. Gillian Hawker) did not advise our full time faculty respondents that we would make their de-identified responses, including their demographic data and responses to open ended questions, accessible through an open access policy. 

We do not have the opportunity to revisit this issue at this point in time as the questionnaire was administered in 2019. Since we told respondents that all data would be confidential, we feel that it would be unethical to provide the study database at this time. However, for individuals who are interested, we would be willing to consider providing access to the data on an individual basis with a written request stating the purpose and how the data would be used.

Individuals interesting in accessing the database may contact Mr. Ed Lorens, 6 Queen’s Park Crescent West, 3rd Floor, Toronto, Ontario, Canada M5S 3H2; E-mail: ed.lorens@utoronto.ca

 Not applicable. Please see above.

'The funders had no role in study design, data collection and analysis, decision to publish, or preparation of the manuscript.'

a. Please clarify the sources of funding (financial or material support) for your study. List the grants or organizations that supported your study, including funding received from your institution.

d. If you did not receive any funding for this study, please state: “The authors received no specific funding for this work.”

 Included in our cover letter. Thank you.

This study was unfunded. The authors received no specific funding for this work.

'Gillian Hawker receives salary support as the Sir John and Lady Eaton Professor and Chair of Medicine, Department of Medicine, University of Toronto. Sharon Straus holds a Tier 1 Canada Research Chair from the Canadian Institutes of Health. Karen Burns holds a Physician Services Incorporated Mid-Career Clinical Research Award. No other competing interests are declared.'

We confirm that this does not alter our adherence to PLOS ONE policies on data sharing and materials. We have added the following,

Gillian Hawker receives salary support as the Sir John and Lady Eaton Professor and Chair of Medicine, Department of Medicine, University of Toronto. Sharon Straus holds a Tier 1 Canada Research Chair from the Canadian Institutes of Health. Karen Burns holds a Physician Services Incorporated Mid-Career Clinical Research Award. No other competing interests are declared. We confirm that salary support and personnel awards did not impact our ability to adhere to PLOS ONE policies on data sharing and materials.

Done.

 The above text regarding potential COIs has been added to the manuscript.

6. PLOS requires an ORCID iD for the corresponding author in Editorial Manager on papers submitted after December 6th, 2016. Please ensure that you have an ORCID iD and that it is validated in Editorial Manager. To do this, go to ‘Update my Information’ (in the upper left-hand corner of the main menu), and click on the Fetch/Validate link next to the ORCID field. This will take you to the ORCID site and allow you to create a new iD or authenticate a pre-existing iD in Editorial Manager. Please see the following video for instructions on linking an ORCID iD to your Editorial Manager account: https://www.youtube.com/watch?v=_xcclfuvtxQ.

I have entered my ORCID ID previously. Verified.

Reviewers' comments:

Reviewer's Responses to Questions

Comments to the Author

1. Is the manuscript technically sound, and do the data support the conclusions?

Reviewer #1: Yes

Reviewer #2: Yes

2. Has the statistical analysis been performed appropriately and rigorously?

Reviewer #1: Yes

Reviewer #2: Yes

3. Have the authors made all data underlying the findings in their manuscript fully available?

Reviewer #1: Yes

Reviewer #2: No

Although, we have no difficulty sharing our database, we did not advise respondents that we would make their de-identified responses, including their demographic data, accessible through an open access policy. We do not have the opportunity to revisit this issue at this point in time. Since we told respondents that all data would be confidential, we feel that it would be unethical to provide the database at this time. However, for individuals who are interested, I would be willing to consider providing access to the data on an individual written request basis.

Please see our response below

4. Is the manuscript presented in an intelligible fashion and written in standard English?

Reviewer #1: Yes

Reviewer #2: Yes

5. Review Comments to the Author

Reviewer #1: It was a pleasure reviewing the manuscript “The impact of organizational culture on professional fulfilment and burnout in an academic Department of Medicine”

The topic is relevant and the idea innovative. Burnout is frequent in the ICU setting and it is largely undiagnosed. The mortality rate of adult ICU patients depends on the severity of illness and patient population; it ranges between 6.4% and 80% during the COVID-19 Pandemic. Interdisciplinary work between many medical professions is essential in an ICU to deliver high quality patient care. Members of the ICU team include physicians, nurses, technicians, therapists, nutritionists, pharmacists, and other support staff making effective management an important element of a highly functioning ICU. Currently, health care delivery has far lower levels of reliability than that achieved in other industries (e.g., aviation) and staff are not supported efficiently. 

Successful improvement of Critical Care requires a perspective that treats the ICU as a complex, socio-technical system and prevents staff to suffer burnout. ICU healthcare workers who provide aggressive care to critical patients have moral distress and are at risk for burnout, which in turn can lead to poor quality patient care and higher job turnover rates (Meltzer et al. Am J Crit Care 2004, Corley et al. J Adv Nurs 2000, Appropricus ESICM Abstract).In another survey 70% of ICU workers perceived conflicts, (Azoulay E, Timsit JF, Sprung CL, et al. Prevalence and Factors of Intensive Care Unit Conflicts: The Conflicus Study. Am J Respir Crit Care Med. 2009 Jul 30m). These were usually considered harmful and were significantly associated with job strain. Workload, communication, and end-of-life care emerged as potential targets for improvement. 

Hamric and colleagues evaluated moral distress in both nurses and doctors. They found that ICU nurses experienced moral distress more often than physicians; however, similar situations provoked moral distress in both groups (Hamric et al. Crit Care Med 2007). Studies related to moral distress and futile or inappropriate care in the ICU do not provide patient-linked data. Consequently, the real extent of the problem of perceived inappropriate care at the end-of-life in the ICU is unknown and the magnitude of situations causing feelings of moral distress may be underestimated. Feelings like frustration, stress, guiltiness, lack of motivation, lack of communication, isolation and finally burn out are common. There is evidence in the medical and non-medical literature suggesting that the burn out leads to low performance and concentration. The performance of all Critical Care workers is prone to error and is often associated with monitoring and daily care tasks like ordering of medication or execution of patient treatment. ERRORS cause distress and can lead to more burnout. A blaming culture can lead to more mistakes and so on. The organisational culture is as complex to measure as is the incidence or severity of burnout. There are few validated tools and these are limited. 

This study is looking at a relevant element and it is using one those few tools that can provide reliable information. The Stanford Professional Fulfilment Index (PFI) reports on professional fulfilment, work exhaustion, interpersonal disengagement.

Major concerns:

The limit of most surveys is the response rate which in this occasion is just above 50%.

The target people for this survey are the faculty members, in reality the entire staff could be included. The ICU is a complex system and isolating one part of the staff can have benefits and disadvantages such as reporting only on one component or one side of the problem.

The percentage of people happy with their condition was high. Results presented in this manuscript would be more valuable if a benchmark was available. This model could be used in several institutions and compared in future.

Thanks for your kind review.

Although higher response rates are achieved in single discipline surveys, a response rate of 50% is considered acceptable for multidisciplinary cross-sectional surveys. Mean response rates of 54% [1] to 61% [2] for physicians and 68%32 for nonphysicians have been reported in recent systematic reviews of postal questionnaires.

References

Asch DA, Jedrzwieski MK, Christakis NA. Response rates to mail surveys published

in medical journals. J Clin Epidemiol 1997;50:1129-36.

Cummings SM, Savitz LA, Konrad TR. Reported response rates to mailed physician

questionnaires. Health Serv Res 2001;35:1347-55.

We agree that all member of individual hospitals or ICUs could be surveyed with our questionnaire. This may represent an area of inquiry for future research.

Our study presents the findings of a biennial survey administered to physicians that are full time faculty members within our DOM. The DOM at the University is the largest DOM in Canada with over 600 full time members including physicians from a wide range of specialties. Most surveys have focused on burnout in multidisciplinary members (nurses, physicians, allied health care providers). Conversely our goal was to examine career satisfaction, professional fulfillment, burnout and the impact of organizational culture on these metrics among physicians from various specialties within our large academic DOM. As such this represents the largest survey ever conducted using the Stanford PFI to measure both professional fulfillment and burnout. This tool was specifically designed to assess these wellness measures in physicians. Selection of this tool, aligned well with our objective which was to examine career satisfaction, professional fulfillment, burnout and the impact of organizational culture on these metrics in physicians. 

Although our findings many not be generalizable to other disciplines, we believe that our findings are generalizable to academic physicians and aligns with our intended goals.

Minor concerns:

-small error in the text and the title page : “The impact of organizational culture on professional fulfillment and burnout in an academic…”

Please change fulfilment

Fulfillment is spelled with two letter l’s throughout.

We checked the Merriam-webster dictionary to verify that this is the correct spelling. 

https://www.merriam-webster.com/dictionary/fulfillment?src=search-dict-box

Definition of fulfillment

1: the act or process of fulfillingthe fulfillment of a promisethe fulfillment of all the requirements

2: the act or process of delivering a product (such as a publication) to a customerthe fulfillment of a book order

Thank you for taking the time to review our manuscript and for your thoughtful input and review.

Reviewer #2: Thank you for the opportunity to review this paper by Dr. Burns and colleagues on the impact of organizational culture on professional fulfillment and burnout in an academic department of medicine. This well-written paper reports the results of a cross-sectional survey of 400 full-time physicians. I have a few comments/queries for the authors to consider:

Abstract:

1) Please provide the score range for the SPFI so the mean scores presented in the results are interpretable

Thank you for this suggestion. The score range has now been added to the abstract.

We used multivariable linear regression to examine the relationship of measures of workplace culture on professional fulfillment and burnout (scores 0-10), controlling for physician factors.

2) Are the "physician factors" referred to on line 40 the same as the "physician characteristic" defined on line 34? if so perhaps change one of the terms for consistency.

These terms represent different facets.

Physician characteristics included age (≤50, 51-60, 61-70, >70 years); gender (female/male); self-reported disability status (yes/no); under-represented minority (URM) membership (yes/no); socioeconomic status (SES) as a child/adolescent (lower/lower-middle/middle/upper-middle/upper); academic rank (lecturer/assistant professor/associate professor/professor), clinical specialty and academic position to determine protected time for scholarly work.

Physician factors include the above physician characteristics and whether respondents were interested in reducing their clinical workload to aid work-life integration (yes/no/not sure).

We have changed physician characteristics to physician factors where appropriate throughout the manuscript. Thank you for noting this.

Introduction

4) The introduction is concise and lays out both the problem and gap well.

Thanks kindly for this comment.

Methods

5) For the assessments, more detail on some of the definitions would be helpful. For example, for self-reported disability status, how was this question asked? Does it include physical, mental, or cognitive disabilities? For participant age, why were those grouping selected? 60% of respondents are in the <50 are group.

Thank you for this suggestion. We have appended the questionnaire as an electronic appendix. (see Appendix 1)

We asked whether individuals perceived that they had a disability.

Whether or not it affects your day-to-day life, are you a person with a disability?

Please check ONE only. 

Response options included Yes, No, Not sure, and Prefer not to answer

We defined disability in the hyperlink as follows:

Disability / Disabilities. A person with a disability is someone who has a long-term or recurring physical, mental, sensory, psychiatric or learning disability and considers oneself to be disadvantaged by reason of that disability, or believes that society is likely to consider them to be disadvantaged by reason of that disability. A person with a disability may also be someone whose functional limitations owing to their disability have been accommodated in their environment. Examples of disabilities include, but are not limited to:

· Addiction to alcohol or drugs

· Chronic illness (e.g. epilepsy, cystic fibrosis, cancer, diabetes)

· Developmental disability (e.g. autism, down syndrome, brain injury) 

· Learning disability (e.g. dyslexia, attention deficit hyperactivity disorder (ADHD))

· Mental illness (e.g. schizophrenia, depression)

· Physical disability (e.g. cerebral palsy, spinal cord injury, amputation)

· Sensory disability (i.e. hearing or vision loss)

Regarding age, we included 4 age categories age (≤50, 51-60, 61-70, >70 years). WE collapsed the lowest three age categories due to small numbers (no one less than 30 years of age, very few between ages 31 and 40, more individuals between 41 and 50 into one category < 50 years. A similar phenomenon occurred at the higher end with only a few physicians >80 years of age. Consequently, we collapsed these individuals with physicians who were > 70 years of age. Looking at he distribution of the data, we felt that the best way to collapse the data was using the aforementioned cut offs as most respondents were between 51 and 70. Thank you for the opportunity to clarify.

Results

6) The sentence starting on line 138 is confusing and it is difficult to tell which numbers apply to which groups given the placement of comas.

Thank you for the opportunity to clarify and remove a comma.

Respondents were similar to non-respondents with respect to age group and hospital affiliation, but were significantly less likely to be a man (60.6% vs. 51.0%), assistant professor (43.6% vs. 38.2%) or cardiologist (14.4% vs. 9.2%).

The revised text reads as follows:

Respondents were similar to non-respondents with respect to age group and hospital affiliation but were significantly less likely to be a man (60.6% vs. 51.0%), assistant professor (43.6% vs. 38.2%) or cardiologist (14.4% vs. 9.2%), respectively.

Discussion

7) The paragraph (starting line 251) discussing the novel findings related to disability should be tempered slightly. With only 12 people in this group the results should be interpreted with caution.

We have changed the wording to acknowledge that this finding should be regarded as hypothesis generating. Thank you for this suggestion.

First sentence: We found a relationship between having a disability and both lower professional fulfillment and higher burnout scores.

Current sentence:

Although limited to 12 respondents, we found a relationship between having a disability and both lower professional fulfillment and higher burnout scores. 

We believe that the remaining text tempers our findings and is aligned with the reviewer’s comments. Thank you.

To our knowledge no prior study has reported these relationships. Unlike gender and URM status, the relationships between disability and both professional fulfillment and burnout were not attenuated by factors related to organizational culture suggesting that the effect of disability on both metrics may be through different mechanisms related to environmental barriers [35] or the effort required by disabled physicians to complete work-related tasks. 

We have also edited the last sentence in this paragraph.

Additional research is needed to confirm these findings in disabled physicians and if true to ascertain how more inclusive work environments can be created for disabled physicians. 

Thank you Dr. Rubulotta and Reviewer #2. We sincerely appreciate you taking the time to review our manuscript.

Not applicable.

---

## [Decision Letter · Decision Letter 1]

24 May 2021

The Impact of Organizational Culture on Professional Fulfillment and Burnout in an Academic Department of Medicine

PONE-D-20-32481R1

Dear Dr. Burns,

We’re pleased to inform you that your manuscript has been judged scientifically suitable for publication and will be formally accepted for publication once it meets all outstanding technical requirements.

Kind regards,

Gerard Hutchinson, MD

Academic Editor

PLOS ONE

Additional Editor Comments (optional):

Reviewers' comments:

Reviewer's Responses to Questions

**Comments to the Author**

1. If the authors have adequately addressed your comments raised in a previous round of review and you feel that this manuscript is now acceptable for publication, you may indicate that here to bypass the “Comments to the Author” section, enter your conflict of interest statement in the “Confidential to Editor” section, and submit your "Accept" recommendation.

Reviewer #1: All comments have been addressed

Reviewer #2: All comments have been addressed

2. Is the manuscript technically sound, and do the data support the conclusions?

Reviewer #1: Yes

Reviewer #2: Yes

3. Has the statistical analysis been performed appropriately and rigorously? 

Reviewer #1: Yes

Reviewer #2: Yes

4. Have the authors made all data underlying the findings in their manuscript fully available?

Reviewer #1: Yes

Reviewer #2: Yes

5. Is the manuscript presented in an intelligible fashion and written in standard English?

Reviewer #1: Yes

Reviewer #2: Yes

6. Review Comments to the Author

Reviewer #1: The manuscript is improved. I am pleased with the current form and I think this is worth publication.

Reviewer #2: Thank you for your careful attention to the reviewer's comments. I have no further comments and look forward to seeing this important work in print.

7. PLOS authors have the option to publish the peer review history of their article (what does this mean?). If published, this will include your full peer review and any attached files.

Reviewer #1: **Yes: **Francesca Rubulotta

Reviewer #2: No

---

## [Editor Report · Acceptance letter]

28 May 2021

PONE-D-20-32481R1 

The Impact of Organizational Culture on Professional Fulfillment and Burnout in an Academic Department of Medicine 

Dear Dr. Burns:

I'm pleased to inform you that your manuscript has been deemed suitable for publication in PLOS ONE. Congratulations! Your manuscript is now with our production department. 

Kind regards, 

on behalf of

Dr. Gerard Hutchinson 

Academic Editor

PLOS ONE